# Segregation Analysis of Rare *NRP1* and *NRP2* Variants in Families with Lymphedema

**DOI:** 10.3390/genes11111361

**Published:** 2020-11-17

**Authors:** Sandro Michelini, Bruno Amato, Maurizio Ricci, Sercan Kenanoglu, Dominika Veselenyiova, Danjela Kurti, Mirko Baglivo, Elena Manara, Munis Dundar, Juraj Krajcovic, Syed Hussain Basha, Sasi Priya, Roberta Serrani, Giacinto A. D. Miggiano, Barbara Aquilanti, Giuseppina Matera, Valeria Velluti, Lucilla Gagliardi, Astrit Dautaj, Matteo Bertelli

**Affiliations:** 1Department of Vascular Rehabilitation, San Giovanni Battista Hospital, 00148 Rome, Italy; s.michelini@acismom.it; 2Department of General and Geriatric Surgery, University of Naples “Federico II”, 80138 Naples, Italy; bruno.amato@unina.it; 3Division of Rehabilitation Medicine, Azienda Ospedaliero-Universitaria, 60126 Ospedali Riuniti di Ancona, Italy; maurizio.ricci@ospedaliriuniti.marche.it (M.R.); roberta.serrani@ospedaliriuniti.marche.it (R.S.); 4MAGI Euregio, 39100 Bolzano, Italy; sercankenan@gmail.com (S.K.); d.veselenyiova@gmail.com (D.V.); genetica.clinica@assomagi.org (D.K.); mirko.baglivo@assomagi.org (M.B.); elena.manara@assomagi.org (E.M.); matteo.bertelli@assomagi.org (M.B.); 5Department of Medical Genetics, Faculty of Medicine, Erciyes University, Kayseri 38039, Turkey; dundar@erciyes.edu.tr; 6Department of Biology, Faculty of Natural Sciences, University of Ss. Cyril and Methodius in Trnava, 91701 Trnava, Slovakia; juraj.krajcovic@ucm.sk; 7MAGI-Balkan, Tirana 1019, Albania; 8Innovative Informatica Technologies, Hyderabad 500 049, India; shb@innovativeinformatica.com (S.H.B.); sasipriya641997@gmail.com (S.P.); 9UOC Nutrizione Clinica, Fondazione Policlinico Universitario A. Gemelli IRCCS, 00168 Rome, Italy; giacintomiggiano@gmail.com (G.A.D.M.); barbara.aquilanti@policlinicogemelli.it (B.A.); giusmatera@yahoo.it (G.M.); valeriavelluti@gmail.com (V.V.); lucilla.gagliardi@gmail.com (L.G.); 10Centro di Ricerche in Nutrizione Umana, Università Cattolica Sacro Cuore, 00168 Rome, Italy; 11EBTNA-Lab, 38068 Rovereto, Italy; 12MAGI’s Lab, 38068 Rovereto, Italy

**Keywords:** NRP1, NRP2, NGS, lymphedema, genetic diagnostics

## Abstract

Neuropilins are transmembrane coreceptors expressed by endothelial cells and neurons. NRP1 and NRP2 bind a variety of ligands, by which they trigger cell signaling, and are important in the development of lymphatic valves and lymphatic capillaries, respectively. This study focuses on identifying rare variants in the *NRP1* and *NRP2* genes that could be linked to the development of lymphatic malformations in patients diagnosed with lymphedema. Two hundred and thirty-five Italian lymphedema patients, who tested negative for variants in known lymphedema genes, were screened for variants in *NRP1* and *NRP2*. Two probands carried variants in *NRP1* and four in *NRP2*. The variants of both genes segregated with lymphedema in familial cases. Although further functional and biochemical studies are needed to clarify their involvement with lymphedema and to associate *NRP1* and *NRP2* with lymphedema, we suggest that it is worthwhile also screening lymphedema patients for these two new candidate genes.

## 1. Introduction

Neuropilins 1 and 2 are transmembrane coreceptor proteins encoded by the *NRP1* and *NRP2* genes. They were first identified in the nervous system of *Xenopus laevis* embryos [1,2]. Although they map to different chromosomes (10p11.22 and 2q33.3, respectively; see Table 1), they have 45–50% amino acid sequence homology [3]. The two neuropilins are implicated in angiogenesis, and both are expressed by endothelial cells and neurons [4].

The two neuropilins have a very short intracellular domain (only about 40 amino acids long) and a longer extracellular region. Both neuropilins complex a large variety of ligands, by which they take part in cell signaling. These transmembrane proteins have a critical role in the regulation of vascular and neural development through binding vascular endothelial growth factors (VEGFs) and semaphorins [5]. The major neuropilin ligands are vascular endothelial growth factors VEGF-A, VEGF-C and VEGF-D [4,6]. While VEGF-A plays a primary role in angiogenesis, VEGF-C and VEGF-D are important for normal development of the lymphatic system. NRP1 and NRP2 also bind other ligands—for instance, hepatocyte growth factor (HGF) [7], EGF receptor [8] and an adhesion receptor L1-CAM [9]. Although they mediate cell signaling by ligand binding, it is unclear whether they also transduce signals on their own. In any case, they take part in many biological processes, and changes in their function may modulate tumor growth and metastasis, affecting vascular and lymphatic biology [10]. This can give rise to pathologies such as cancer and autoimmune disorders [11].

While NRP1 is predominantly expressed in the arterial endothelium and lymphatic valves, NRP2 mainly functions in venous and lymphatic endothelial cells (LECs) [12]. This indicates that NRP1 acts as a semaphorin receptor in the valve endothelium throughout lymphatic vessel development [12]. In contrast, NRP2 is implicated in VEGF-C–driven lymphatic vessel growth [13], while double-heterozygous *nrp2^+/−^ vegfr3^+/−^* mice show a decrease in lymphatic vessel sprouting in adult organs (Table 1) [13].

Most information about neuropilin function comes from studies with mice. Early studies of *Nrp1*^−/−^ mice showed deficient neuronal vascularization, aortic arch malformations and impaired yolk sac vascularization [14,15]. Abnormal blood vessel formation due to overexpression of the *Nrp1* gene was also observed [16], although the authors of the study did not investigate the lymphatic phenotype.

*Nrp1* was recently implicated in lymphatic valve development via Sema3a/Nrp1 signaling [12]. Sema3a is a ligand bound by Nrp1. Deficiency or blockage of Nrp1 bound to Sema3a leads to malformations of a specific type of LEC that later migrates from the wall of lymphatic vessels to give rise to lymphatic valves. Defects in this process lead to abnormal lymphatic valve development [13,17,18].

In comparison, the *Nrp2*^−/−^ mouse model provided evidence of the selective role of Nrp2 in the development of lymphatic vessels. Nrp2-deficient mice showed significantly fewer or a complete absence of small lymphatic vessels and capillaries in many organs. The small number of vessels that did develop were wrongly positioned, and in some cases, small vessels and capillaries were larger in size. A decrease in LEC proliferation was also observed, indicating a role of Nrp2 in the proliferation of LECs in selected types of lymphatic vessels and implying the control of positional guidance by Nrp2 [19].

Knocking out *Nrp1* and *Nrp2* separately has evident effects on the mouse phenotype, and the joint deletion of *Nrp*1 and *Nrp2* has much graver consequences. Takashima et al. [20] reported that *Nrp1^−/−^/Nrp2^−/−^* mice died at an early stage of embryogenesis (E8.5) and displayed severely defective blood vessel development in the yolk sac. In the study, the authors observed that double-knockout mice had severer vascular malformations than *Nrp1-* or *Nrp2-*deleted animals. Surprisingly, heterozygous mice with *Nrp1^+/−^/Nrp2^−/−^* or *Nrp1^−/−^/Nrp2^+/^*^−^ genotypes did not survive embryo development either but lived longer than double-knockout mice (E10–E10.5) [20]. Although the authors did not study the development of lymphatics in these double-knockout mice, the premature death of the embryos and the fact that *Nrp1* and *Nrp2* deletions caused severe lymphatic system malformations suggested that a double knockout also results in impaired lymphatic system development.

The development of small lymphatic vessels, capillaries and valves is important, and its impairment may lead to an overaccumulation of lymph in the tissues and lymphedema. Lymphedema is a pathological condition characterized by swelling, usually at the extremities, accompanied by pain and inflammation [21]. Lymphatic capillaries, the development of which is regulated by *NRP2,* are indispensable for normal drainage of interstitial fluid into the lymphatic system. Lymphatic capillaries lack a basement membrane and feature endothelial cell junctions that grant the passage of interstitial fluid into the lymphatic system [22]. When lymphatic capillaries and small vessels are malformed, interstitial fluid cannot enter lymphatic collecting vessels and accumulates in the tissues. The proper functioning of valves is also crucial to ensure retrograde lymph flow. Valve defects cause lymph flow obstruction and fluid accumulation [22]. The blockage of lymph flow and lymph accumulation are the primary causes of lymphedema.

In our study, a cohort of 235 Italian lymphedema patients, who tested negative for variants in known lymphedema-associated genes, was screened by a next-generation sequencing (NGS)-targeted panel for variants the in *NRP1* and *NRP2* genes. A segregation analysis in familial lymphedema cases added evidence that *NRP1* and *NRP2* qualify as candidate genes to include in the genetic testing of lymphedema patients.

## 2. Materials and Methods

### 2.1. Clinical Evaluation

We retrospectively enrolled 235 Caucasian lymphedema patients in this study, which was conducted in accordance with the Declaration of Helsinki, and whose protocol was approved by the Ethics Committee of Azienda Sanitaria dell’Alto Adige, Italy (Approval No. 94-2016). No consanguinity was reported in any family. Clinical diagnosis of lymphedema was made in hospitals across Italy, according to generally accepted criteria. All subjects gave their informed consent for inclusion before they participated in the study.

### 2.2. Genetic Analysis

Genetic testing was performed on genomic DNA extracted from saliva or peripheral blood of probands, as previously described [23]. Briefly, a custom-made oligonucleotide probe library was designed to capture all coding exons and flanking exon/intron boundaries (±15 bp) of genes known to be associated with lymphedema [24]. We added the candidate genes *NRP1* and *NRP2* to our panel. DNA from probands was analyzed for germline variants. Variants were confirmed by bidirectional Sanger sequencing on a CEQ8800 Sequencer (Beckman Coulter, Brea, CA, USA).

We searched the international Single Nucleotide Polymorphism Database (dbSNP) and the Human Gene Mutation Database professional for all nucleotide changes. In-silico evaluation of the pathogenicity of sequence changes in NRP1 and NRP2 was performed using the Variant Effect Predictor tool and MutationTaster. Minor allele frequencies were checked in the Genome Aggregation Database (gnomAD, https://gnomad.broadinstitute.org/). All variants were evaluated according to the American College of Medical Genetics and Genomics guidelines [25]. Detailed pretest genetic counseling was provided to all subjects, who were then invited to sign informed consent to use of their anonymized genetic results for research.

### 2.3. In-Silico Analysis

The primary amino acid sequences of NRP1 and NRP2 in FASTA format (Table 2) were used as targets to search template libraries (i.e., the SWISS-MODEL Template Library (version 2019-10-24) and the Protein Data Bank (release 2019-10-18)) [26] for matching evolution-related structures by means of BLAST [27] and HHBlits [28], as previously described [23]. Briefly, models were based on target-template alignment using ProMod3 of the SWISS-MODEL server [29]. An alternative model was obtained with ProMod-II in those cases in which loop modeling failed with ProMod3 [30]. Coordinates conserved between the target and the template were copied from the template to the model. Insertions and deletions were remodeled using a fragment library. Side chains were then rebuilt. Finally, the geometry of the resulting model was regularized with the CHARMM27 force field [31]. Global and per-residue model quality was assessed using the QMEAN scoring function [32]. BioVia Discovery Studio Visualizer (version 17.2) [33] was used to visualize the modeled protein, to vary the targeted amino acids and to analyze interactions at the molecular level.

## 3. Results

### 3.1. Clinical and Genetic Evaluation

DNA from 235 Caucasian patients diagnosed with lymphedema, from different Italian hospitals, previously testing negative for variants in known lymphedema and lymphatic malformation-causing genes, was screened for variants in new candidate genes. Here, we report the results for *NRP1* and *NRP2*. No consanguinity was reported in their families.

NGS screening detected two different variants in NRP1 in two families and four different variants in NRP2 in four families. All the variants were heterozygous. The clinical features of the probands and family members are shown in Table 3. A segregation analysis performed on family members carrying variants in *NRP1* are shown in Figure 1. The pedigrees of families with variants in NRP2 are shown in Figure 2.

Regarding variants in the *NRP1* gene, the first proband, male, 68 years, has had lymphedema of the lower limbs since his youth. The case is sporadic, and no other family members of the proband were tested. The proband carries a missense single-nucleotide variation c.2557A > C (dbSNP rs760388137) with a GnomAD-reported frequency of 0.0001178.

The second proband with an *NRP1* variant, female, 54 years, has stage 2 lymphedema of the left foot and ankle, diagnosed at age 27. The case is familial: the proband’s daughter also has stage 2 lymphedema of the lower limbs (Figure 1). Both carry the same heterozygous missense variant c.1655G > A (dbSNP ID rs757990959) with a frequency of 0.00004873 (GnomAD) classified variant of unknown significance (VUS) by the ACMG [34].

Regarding variants in the *NRP2* gene, we identified four families with rare variants. In the first family, the proband, female, 18 years, has congenital edema of the left hand (lymphedema stage 2). Indeed, upper limb lymphoscintigraphy failed to visualize the left axillary lymph nodes. She carries a missense variant c.580T > C (rs755679361), which has an allele frequency in healthy control populations of 0.000008122, according to GnomAD. The segregation analysis in available family members showed that the proband inherited the variant from her father. The father was apparently healthy; however, lymphoscintigraphy showed clear bilateral lymph transport problems, especially on the left side (Figure 3).

The second proband with an *NRP2* variant, male, nine years, developed edema of the left foot, ankle and heel after vaccination at age three months. The case is sporadic, and no other family members were tested. The missense variant is classified as VUS (c.1748T > C) (dbSNP rs746130411), with a GnomAD frequency of 0.000004068.

The third proband, female, 45 years, was diagnosed with edema of the right foot at age 32. The heterozygous missense variant is c.838C >T (dbSNP rs79750907) and has a frequency of 0.001783 (GnomAD). This is a familial case, since the proband’s son was also diagnosed with left foot lymphedema at age nine. The son was found to carry the same variant as the proband.

The fourth proband, female, 50 years, was diagnosed with swelling of the lower limbs at age 42. She carries a heterozygous single-nucleotide missense variant c.1000C > T (dbSNP rs114144673), with a frequency of 0.001525 (GnomAD). This is a sporadic case without family history, and no other family members were tested.

### 3.2. In Silico Analysis, Template Selection and Building

In an attempt to study the possible role of the variants found in our cohort compared to the wildtype, we performed an in-silico study. A wildtype template (Table 2) search with BLAST and HHBlits against the SWISS-MODEL template library (last update: 2019-10-24; last included PDB release: 2019-10-18) produced a total of 267 and 254 templates that matched the *NRP1* and *NRP2* genes, respectively, with different sequence identity and quality percentages. Details of the top 10 templates are shown in Table 4.

Based on the percentage of sequence identity, similarity and best quality square, the 4gz9.1.A and 2qql.1.A chains were selected to align the template and query sequences in order to build models of NRP1 and NRP2 (Figure 4 and Figure 5).

Then, we entered the model in the Discovery studio visualizer to generate versions of the NRP1 structure with Arg552Gln and versions of the NRP2 structure with Phe194Val, Pro280Ser, Arg334Cys and Ile583Thr. Unfortunately, the template to generate the model for Ile853 was not available, so we could not study its interaction with adjacent residues. A molecular-level interaction analysis between native/variant residues was performed (Figure 6, Figure 7, Figure 8, Figure 9 and Figure 10 show snapshots). Details of the residues involved in the interactions, the types of bond they formed and bond lengths in angstrom units are shown in Table 5 and Table 6.

Briefly, for all the residues analyzed, there is a change in the number or length of the bonds that are formed in the variant with respect to the modeled wildtype structure. Indeed, the in-silico analysis showed that NRP1 encoded with Arg552 has slightly different stability from the Gln552 variant, because it has seven interactions with adjacent residues, whereas the variant has only three that exactly match three of the Arg552 interactions (Table 5 and Figure 6).

By contrast, NRP2 encoded with Phe194 has slightly different stability from the Val194 variant due to four interactions with adjacent residues, whereas the variant has only two, similar to Phe194 but slightly different in bond length (Table 6 and Figure 7).

In the case of Pro280Ser, there is a shift from a nonpolar residue (Pro) to a polar residue (Ser). Differences in the amino acid side chain greatly alter the interaction of the residues with the phenylalanine at residue 425, leading to a loss of Pi interaction. These results suggest that the overall protein structure is altered by these different interactions with nearby residues and is functionally defective (Table 6 and Figure 8).

In the case of Arg334Cys, Arg334 has six interactions with adjacent residues, whereas Cys334 has only three similar to those of Arg334, and the interaction with Ile226 has a different hydrophobic interaction bond length (Table 6 and Figure 9).

Likewise, in the case of Ile583Thr, Ile583 has six interactions with adjacent residues, whereas the Thr583 variant has only four that are the same, but the Pi interaction with Trp578 varies slightly in bond length (Table 6 and Figure 10).

## 4. Discussion

Lymphedema is a progressive disease caused by lymphatic system malformations and impaired lymph flow. A panel consisting of 29 known lymphedema genes is currently used for the genetic testing of patients [24] but only detects relevant variants in 25–30% of patients [35,36]. This highlights the need for new target genes to screen. Here, we report our findings from screening a cohort of 235 patients, negative for variants in the said 29 genes, for variants in the neuropilin-coding genes *NRP1* and *NRP2.*

Neuropilins are highly conserved transmembrane receptors that perform various functions in vessels, neurons and tumors, because they bind structurally to different ligands and coreceptors. Although NRP1 and NRP2 share basic structural features, they are implicated in different biological processes, such as the development of blood and lymphatic endothelial cells [37]. They have a role in the development of lymphatic valves [13,17,18], lymphatic vessels and capillaries [12,19,22].

Our screening detected two (2/235; 0.852%) and four (4/235; 1.702%) distinct probands with missense variants in *NRP1* and *NRP2,* respectively. In four out of six cases, the proband was a female.

Overall, our results showed rare variants in *NRP1* and *NRP2* in patients who tested negative for variants in the 29 genes already associated with lymphedema. In three out of six cases, family members were screened, highlighting segregation of the variant with lymphedema or with subclinical signs of abnormal lymph flow. Bioinformatic modeling added evidence of a possible impact of the variant on the stability of the wildtype protein. Further biochemical studying or in-vitro studying is necessary to verify the altered function of coreceptors carrying the variant. In addition, other studies aiming at evaluating the effect of the interaction of the altered NRP1 with its principal ligand SEMA3A, or of the altered NRP2 with its coreceptor VEGFR3, could be important to define whether the variants also have a role in such signaling. Moreover, the loss of SEMA3A function by the specific inhibition of its interaction with NRP1 leads to impairment of lymphatic valve development [12], while variants in VEGFR3, a tyrosine kinase receptor important in the development and establishment of the lymphatic system, are known to cause Milroy disease, a form of primary lymphedema [38].

Additional studies could shed light on altered pathways, leading to a better understanding of the mechanism of action of the alteration; in particular, it could help determine whether the phenotype seen in our patients is due to haploinsufficiency or to a dominant negative effect of the coreceptor carrying the variant.

Although these results are not sufficient to associate the two genes, known to be involved in the development of lymphatic capillaries and valves, with lymphedema, they suggest that it would be worthwhile screening a larger cohort of patients for variants in *NRP1* and *NRP2,* especially in large families with more than one affected subject, and performing a further functional analysis of the variants identified by us in in vitro and in vivo models to confirm their involvement in the development of lymphedema.

## 5. Conclusions

A comprehensive review of the literature strongly suggests that neuropilins NRP1 and NRP2 play a role in the development of lymphatic capillaries and lymphatic valves, crucial components of the lymphatic system. If these components do not function normally, the lymph does not drain, and fluid may accumulate in tissues, eventually leading to lymphedema. Given their functional role and our findings, we suggest that *NRP1* and *NRP2* should be considered candidate genes for inclusion in the gene panel for genetic testing of lymphedema patients.

## Figures and Tables

**Figure 1 genes-11-01361-f001:**
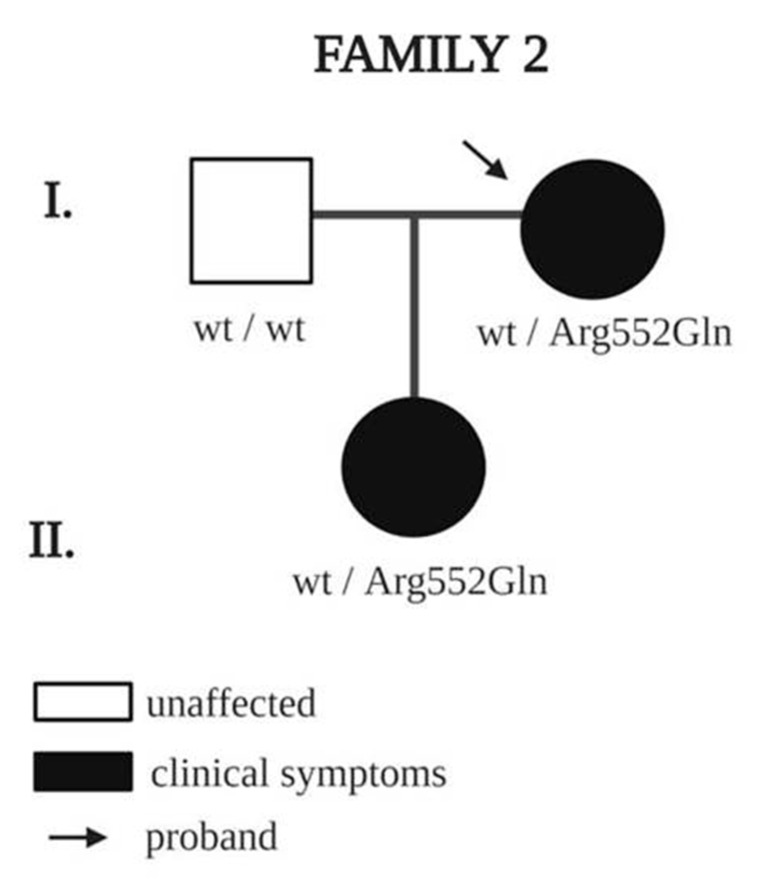
Pedigree of family 2 with a rare *NRP1* variant. Square indicates male, and circles indicate females; black indicates clinical symptoms of lymphedema. Arrows indicate probands, and wt indicates wildtype amino acid.

**Figure 2 genes-11-01361-f002:**
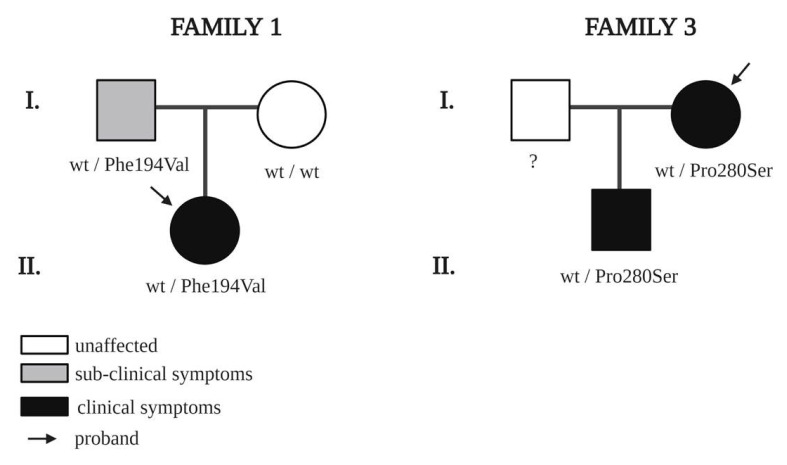
Pedigrees of families with rare *NRP2* variants. Squares indicate males and circles, females; black indicates clinical symptoms of lymphedema; grey indicates subclinical symptoms confirmed by lymphoscintigraphy. Arrows indicate probands. ? means “not screened”, and wt indicates wildtype amino acid.

**Figure 3 genes-11-01361-f003:**
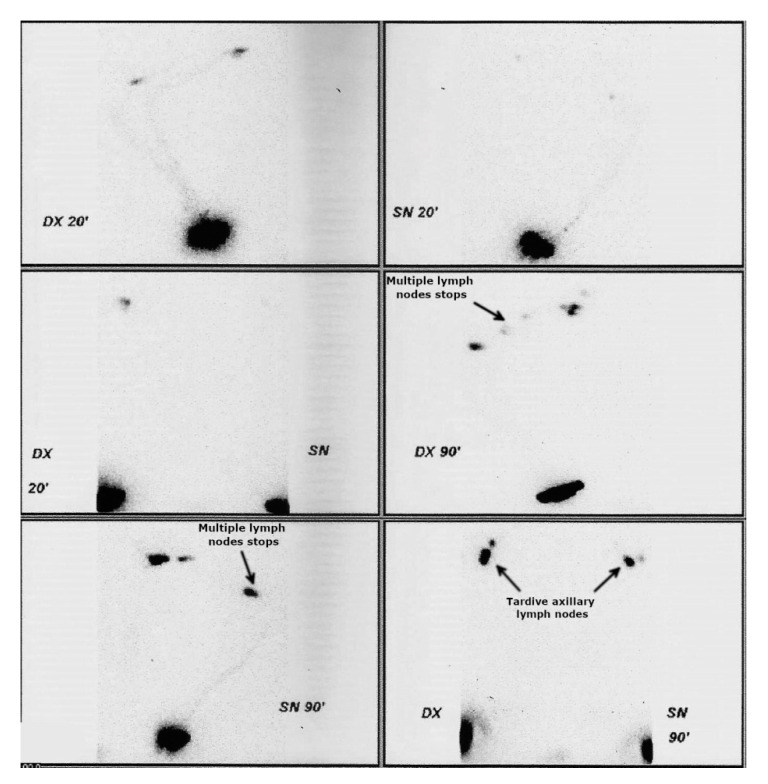
Lymphoscintigraphy of father of the proband with c.580T > C variants in *NRP2* (family 1). Although phenotypically healthy, lymphoscintigraphy highlighted subclinical anomalies (arrows): radiotracer accumulates in multiple lymph nodes of the left (panel 4) and right arm (panel 5). After 90 min of tracer administration, there is reduced tardive visualization of the right axillary lymph nodes (right axillary packet hypogenesis) and even more reduced tardive visualization of the left axillary lymph nodes (left axillary packet hypogenesis) (arrows in panel 6).

**Figure 4 genes-11-01361-f004:**
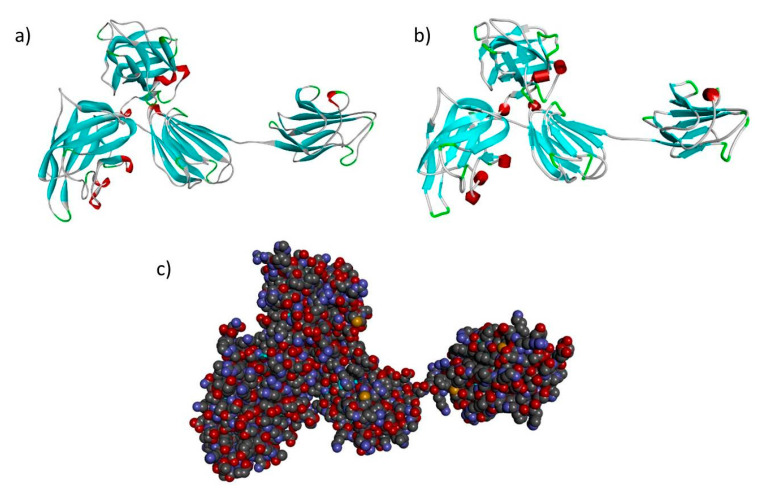
Modeled structure of the *NRP1* gene in (**a**) ribbon (**b**) schematic and (**c**) CPK views. Cyan regions are β sheets, white represents loops and red represents α helices.

**Figure 5 genes-11-01361-f005:**
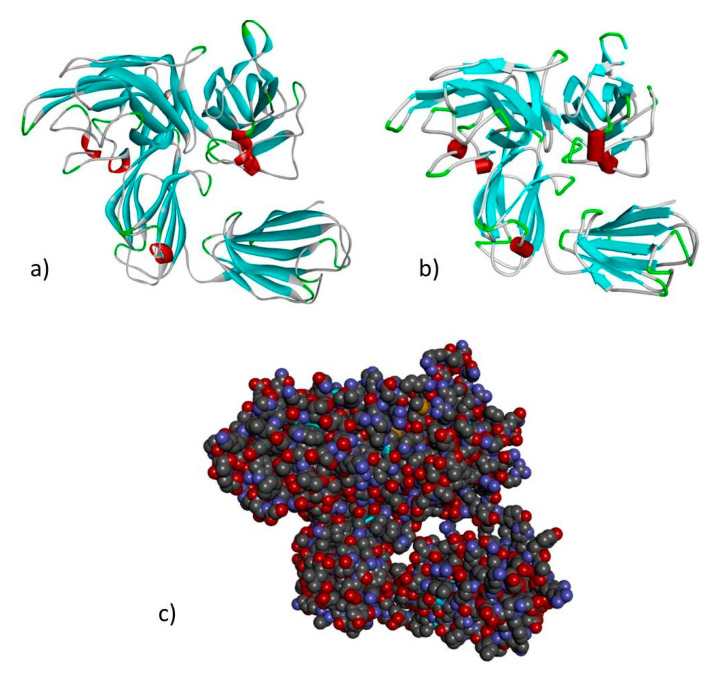
Modeled structure of the *NRP2* gene in (**a**) ribbon (**b**) schematic and (**c**) CPK views. Cyan regions are β sheets, white represents loops and red represents α helices.

**Figure 6 genes-11-01361-f006:**
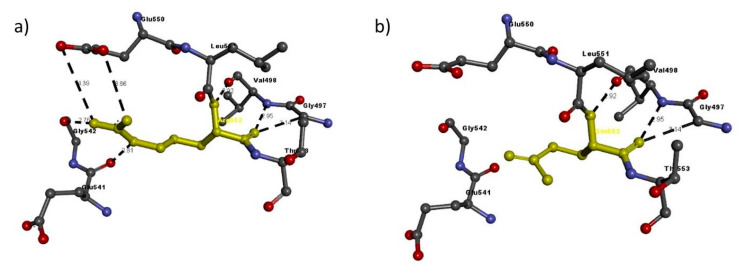
Molecular interactions of (**a**) Arg552 and (**b**) Gln552 (in yellow) of the modeled NRP1 protein with adjacent residues.

**Figure 7 genes-11-01361-f007:**
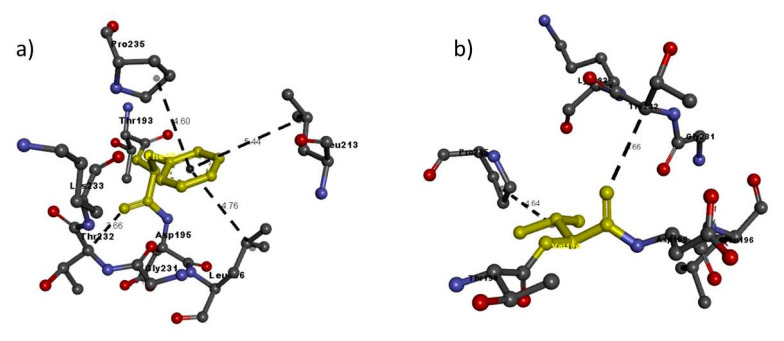
Molecular interactions of (**a**) Phe194 and (**b**) Val194 (in yellow) of the modeled NRP2 protein with adjacent residues.

**Figure 8 genes-11-01361-f008:**
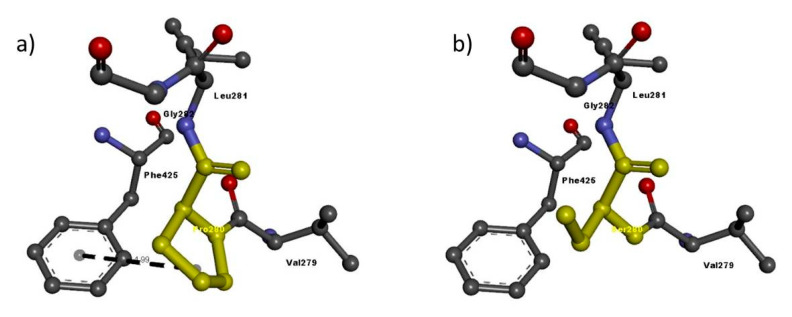
Molecular interactions of (**a**) Pro280 and (**b**) Ser280 (in yellow) of the modeled NRP2 protein with adjacent residues.

**Figure 9 genes-11-01361-f009:**
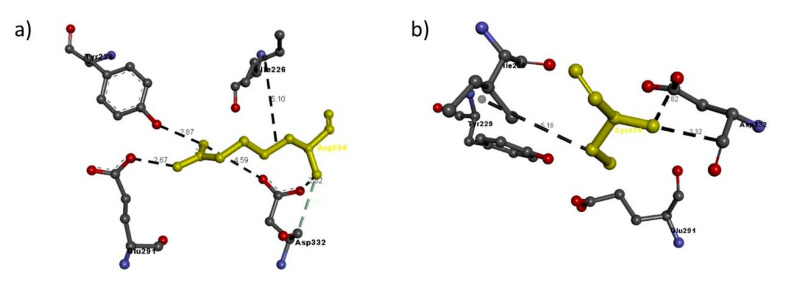
Molecular interactions of (**a**) Arg334 and (**b**) Cys334 (in yellow) of the modeled NRP2 protein with adjacent residues.

**Figure 10 genes-11-01361-f010:**
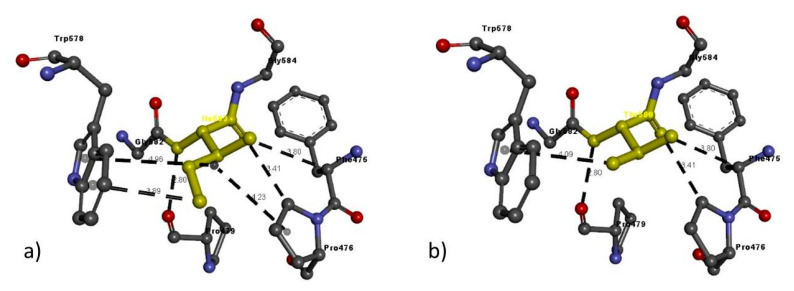
Molecular interactions of (**a**) Ile583 and (**b**) Thr583 (in yellow) of the modeled NRP2 protein with adjacent residues.

**Table 1 genes-11-01361-t001:** Comparison of *NRP1* and *NRP2.*

Characteristics	*NRP1*	*NRP2*
Localization	10p11.22	2q33.3
Gene function	Protein coding	Protein coding
Expression in lymphatic system	Predominantly lymphatic valves	Small lymphatic vessels and capillaries
Lethality in mouse model	Embryo death at E8.5 (*Nrp1^−/−^/Nrp2^−/−^*) Embryo death at E10-E10.5 (*Nrp1^+/−^/Nrp2^−/−^; Nrp1^−/−^/Nrp2^+/−^*)	
Lymphatic phenotype in mouse model	*Nrp1*^−/−^ malformation of valve precursor LECs [18]	*Nrp2*^−/−^ reduction or loss of small lymphatic vessels and capillaries [19] Nrp2+/− and Vegfr3+/− abnormal lymphatic development and reduced lymphatic vessel branching [13]

**Table 2 genes-11-01361-t002:** NRP1 and NRP2 primary amino acid sequences. Primary amino acid sequences used to search for templates to build models for NRP1 and NRP2 proteins.

NRP1	MERGLPLLCAVLALVLAPAGAFRNDKCGDTIKIESPGYLTSPGYPHSYHPSEKCEWLIQAPDPYQRIMINFNPHFDLEDRDCKYDYVEVFDGENENGHFRGKFCGKIAPPPVVSSGPFLFIKFVSDYETHGAGFSIRYEIFKRGPECSQNYTTPSGVIKSPGFPEKYPNSLECTYIVFVPKMSEIILEFESFDLEPDSNPPGGMFCRYDRLEIWDGFPDVGPHIGRYCGQKTPGRIRSSSGILSMVFYTDSAIAKEGFSANYSVLQSSVSEDFKCMEALGMESGEIHSDQITASSQYSTNWSAERSRLNYPENGWTPGEDSYREWIQVDLGLLRFVTAVGTQGAISKETKKKYYVKTYKIDVSSNGEDWITIKEGNKPVLFQGNTNPTDVVVAVFPKPLITRFVRIKPATWETGISMRFEVYGCKITDYPCSGMLGMVSGLISDSQITSSNQGDRNWMPENIRLVTSRSGWALPPAPHSYINEWLQIDLGEEKIVRGIIIQGGKHRENKVFMRKFKIGYSNNGSDWKMIMDDSKRKAKSFEGNNNYDTPELRTFPALSTRFIRIYPERATHGGLGLRMELLGCEVEAPTAGPTTPNGNLVDECDDDQANCHSGTGDDFQLTGGTTVLATEKPTVIDSTIQSEFPTYGFNCEFGWGSHKTFCHWEHDNHVQLKWSVLTSKTGPIQDHTGDGNFIYSQADENQKGKVARLVSPVVYSQNSAHCMTFWYHMSGSHVGTLRVKLRYQKPEEYDQLVWMAIGHQGDHWKEGRVLLHKSLKLYQVIFEGEIGKGNLGGIAVDDISINNHISQEDCAKPADLDKKNPEIKIDETGSTPGYEGEGEGDKNISRKPGNVLKTLDPILITIIAMSALGVLLGAVCGVVLYCACWHNGMSERNLSALENYNFELVDGVKLKKDKLNTQSTYSEA
NRP2	MDMFPLTWVFLALYFSRHQVRGQPDPPCGGRLNSKDAGYITSPGYPQDYPSHQNCEWIVYAPEPNQKIVLNFNPHFEIEKHDCKYDFIEIRDGDSESADLLGKHCGNIAPPTIISSGSMLYIKFTSDYARQGAGFSLRYEIFKTGSEDCSKNFTSPNGTIESPGFPEKYPHNLDCTFTILAKPKMEIILQFLIFDLEHDPLQVGEGDCKYDWLDIWDGIPHVGPLIGKYCGTKTPSELRSSTGILSLTFHTDMAVAKDGFSARYYLVHQEPLENFQCNVPLGMESGRIANEQISASSTYSDGRWTPQQSRLHGDDNGWTPNLDSNKEYLQVDLRFLTMLTAIATQGAISRETQNGYYVKSYKLEVSTNGEDWMVYRHGKNHKVFQANNDATEVVLNKLHAPLLTRFVRIRPQTWHSGIALRLELFGCRVTDAPCSNMLGMLSGLIADSQISASSTQEYLWSPSAARLVSSRSGWFPRIPQAQPGEEWLQVDLGTPKTVKGVIIQGARGGDSITAVEARAFVRKFKVSYSLNGKDWEYIQDPRTQQPKLFEGNMHYDTPDIRRFDPIPAQYVRVYPERWSPAGIGMRLEVLGCDWTDSKPTVETLGPTVKSEETTTPYPTEEEATECGENCSFEDDKDLQLPSGFNCNFDFLEEPCGWMYDHAKWLRTTWASSSSPNDRTFPDDRNFLRLQSDSQREGQYARLISPPVHLPRSPVCMEFQYQATGGRGVALQVVREASQESKLLWVIREDQGGEWKHGRIILPSYDMEYQIVFEGVIGKGRSGEIAIDDIRISTDVPLENCMEPISAFAGENFKVDIPEIHEREGYEDEIDDEYEVDWSNSSSATSGSGAPSTDKEKSWLYTLDPILITIIAMSSLGVLLGATCAGLLLYCTCSYSGLSSRSCTTLENYNFELYDGLKHKVKMNHQKCCSEA

**Table 3 genes-11-01361-t003:** Clinical features of probands (grey background) and family members with variants in the *NRP1* and *NRP2* genes. Abbreviations: M = male, F = female and ACMG = American College of Medical Genetics and Genomics. VUS: variant of unknown significance.

Gene	Family	Pedigree	Sex	Age	Clinical Features	Age of Onset	Familial Case	Variant Nomenclature (NRP1: NM_003873.6; NP_001019799.1; NRP2: NM_003872.2; NP_003863.2)	dbSNP id	ACMG Classification	Frequency in GnomAD
NRP1	1	Proband	M	68	lymphedema of the lower limbs	Juvenile	NO	c.2557A > C; p.Ile853Leu	rs760388137	Likely benign	0.0001178
NRP1	2	Proband	F	54	edema of left foot and ankle	27	YES	c.1655G > A; p.Arg552Gln	rs757990959	VUS	0.00004873
NRP1	2	Daughter	F	21	lymphedema of left lower limb	10	YES	c.1655G > A; p.Arg552Gln			
NRP2	1	Proband	F	18	edema of left hand	Congenital	NO	c.580T > G; p.Phe194Val	rs755679361	VUS	0.000008122
NRP2	1	Mother	F	44	Healthy	/	NO	-			
NRP2	1	Father	M	46	Apparently healthy but with subclinical symptoms	/	NO	c.580T > G; p.Phe194Val			
NRP2	2	Proband	M	9	edema of left foot, ankle and heel since vaccination	3 months	NO	c.1748T > C; p.Ile583Thr	rs746130411	VUS	0.000004068
NRP2	3	Proband	F	45	edema of right foot	32	YES	c.838C > T; p.Pro280Ser	rs79750907	Likely benign	0.001783
NRP2	3	Son	M	15	edema of left foot	9	YES	c.838C > T; p.Pro280Ser			
NRP2	4	Proband	F	50	swelling of lower limbs	42	NO	c.1000C > T; p.Arg334Cys	rs114144673	Likely benign	0.001525

**Table 4 genes-11-01361-t004:** Best models for 3D modeling of the NRP1 and NRP2 structures. Abbreviation: QSQE= quaternary structure quality estimate.

	Template	Seq Identity	Oligo State	QSQE	Found by	Method	Resolution	Seq Similarity	Coverage	Description
NRP1	4gz9.1.A	91.50	monomer	-	BLAST	X-ray	2.70Å	0.60	0.61	Neuropilin-1
	4gz9.1.A	91.50	monomer	-	HHBlits	X-ray	2.70Å	0.60	0.61	Neuropilin-1
	2qql.1.A	51.95	Homodimer	0.37	HHBlits	X-ray	3.10Å	0.46	0.61	Neuropilin-2
	2qql.1.A	53.14	Homodimer	0.35	BLAST	X-ray	3.10Å	0.47	0.60	Neuropilin-2
	2qqk.1.A	51.95	Monomer	-	HHBlits	X-ray	2.75Å	0.48	0.61	Neuropilin-2
	2qqk.1.A	53.14	Monomer	-	BLAST	X-ray	2.75Å	0.47	0.60	Neuropilin-2
	2qqm.1.A	99.78	monomer	-	BLAST	X-ray	2.00Å	0.62	0.48	Neuropilin-1
	2qqm.1.A	99.78	monomer	-	HHBlits	X-ray	2.00Å	0.62	0.48	Neuropilin-1
	2qqo.1.A	51.36	monomer	-	HHBlits	X-ray	2.30Å	0.46	0.48	Neuropilin-2
	2qqo.1.A	52.51	monomer	-	BLAST	X-ray	2.30Å	0.46	0.47	Neuropilin-2
NRP2	2qql.1.A	100.00	Homo-monomer	0.51	BLAST	X-ray	3.10 Å	0.62	0.62	Neuropilin-2
	2qqk.1.A	100.00	monomer	-	BLAST	X-ray	2.75Å	0.62	0.62	Neuropilin-2
	2qql.1.A	100.00	Homodimer	0.51	HHBlits	X-ray	3.10Å	0.62	0.61	Neuropilin-2
	2qqk.1.A	100.00	monomer	-	HHBlits	X-ray	2.75Å	0.62	0.61	Neuropilin-2
	2qqo.1.A	100.00	Monomer	-	HHBlits	X-ray	2.30Å	0.62	0.48	Neuropilin-2
	2qqm.1.A	51.81	Monomer	-	HHBlits	X-ray	2.00Å	0.46	0.47	Neuropilin-1
	2qqm.1.A	53.05	monomer	-	BLAST	X-ray	2.00Å	0.46	0.48	Neuropilin-1
	2qqj.1.A	100.00	monomer	-	BLAST	X-ray	1.95Å	0.62	0.34	Neuropilin-2
	2qqj.1.A	100.00	monomer	-	HHBlits	X-ray	1.95Å	0.62	0.34	Neuropilin-2
	2qqi.1.A	51.77	monomer	-	BLAST	X-ray	1.80Å	0.46	0.33	Neuropilin-1

**Table 5 genes-11-01361-t005:** Details of molecular interactions with adjacent residues of wildtype Arg552 or variant Gln552 of the modeled NRP1 protein in family 2.

	Variant	Amino Acid	Molecular Interactions	Bond Length in Angstroms	Bond Type
NRP1 FAM2	Arg552Gln	Arg552	Arg552: N–Glu550: O	3.66	H-bond
			Arg552: N–Glu550: O	4.39	H-bond
			Val498: N–Arg552: O	2.95	H-bond
			Arg552: N–Val498: O	2.92	H-bond
			Arg552: N–Glu541: O	2.81	H-bond
			Arg552: N–Gly542: O	2.76	H-bond
			Gly497: C–Arg552: O	3.14	H-bond
		Gln552	Val498: N–Gln552: O	2.95	H-bond
			Gln552: N–Val498: O	2.92	H-bond
			Gly497: C–Gln552: O	3.14	H-bond

**Table 6 genes-11-01361-t006:** Details of molecular interactions with adjacent residues of the wildtype or variant residue in the modeled NRP2 protein in family 1 (Phe194Val), family 3 (Pro280Ser), family 4 (Arg334Cys) and family 2 (Ile583Thr).

	Variant	Amino Acid	Molecular Interactions	Bond Length in Angstroms	Bond Type
NRP2 FAM 1	Phe194Val	Phe194	Thr232: C–Phe194: O	3.66	H-bond
			Phe194: A–Phe196	4.76	H-bond
			Phe1941: A–Phe213	5.44	Pi interaction
			Phe194: A–Pro235	4.60	Pi interaction
		Val194	Thr232: C–Val194: O	3.66	H-bond
			Val194: A–Pro235	4.64	Hydrophobic interaction
NRP2 FAM 3	Pro280Ser	Pro280	Phe425–Pro280	4.99	Pi interaction
		Ser280	No bonds	-	-
NRP2 FAM 4	Arg334Cys	Arg334	Arg334: N–Glu291: O	2.67	H-bond
			Arg334: N–Asp332: O	4.59	H-bond
			Arg334: N–Asp332: O	2.82	H-bond
			Arg334: N–Tyr229: O	2.87	H-bond
			Asp332: C–Arg334: N	3.32	H-bond
			Arg334–Ile226	5.10	Hydrophobic interaction
		Cys334	Cys334: N - Asp332: O	2.82	H-bond
			Asp332: C–A:Cys334:N	3.32	H-bond
			A:Cys334–A:Ile226	5.18	Hydrophobic interaction
NRP2 FAM 2	Ile583Thr	Ile583	Ile583: N–Pro479: O	2.80	H-bond
			Phe475: C–Ile583: O	3.80	H-bond
			Pro476: C–Ile583: O	3.41	H-bond
			Ile583: C–Trp578	3.89	Pi interactions
			Pro476: A–Ile583	4.23	Pi interactions
			Trp578: A–Ile583	4.91	Pi interactions
		Thr583	Thr583: N–Pro479: O	2.80	H-bond
			Phe475: C–Thr583: O	3.80	H-bond
			Pro476: C–Thr583: O	3.41	H-bond
			Thr583: O–Trp578	4.09	Pi interactions

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
