# Peer review of "Segregation Analysis of Rare NRP1 and NRP2 Variants in Families with Lymphedema"

_genes, 2020, doi:10.3390/genes11111361_

Round 1
Reviewer 1 Report
The authors used NSG approach to investigate variants in Italian lymphedema patients. Several rare variants in the NRP1 and NRP2 genes have been identified, and modeled protein structures show changes of molecular interaction in those variants. Overall, the manuscript is well-written and the data is clearly presented. As the author mentioned, it is unclear whether NRP1 and NRP2 transduce signals on their own, but clearly they partner with Vegfr2 and Vegfr3 to regulate Vegf-c signaling in lymphatic endothelial cell. It would be informative to show if those variants changes NRP1 and NRP2 interaction with VEGFR2 and VEGFR3.
Author Response
Dear reviewer,
Thank you for your review. The information requested could certainly be very informative and could improve the manuscript, but at the present time, we are unable to obtain these results. In response, we modified the Discussion, mentioning this analysis as future research to perform.
Best regards
Reviewer 2 Report
In this work, Michelini et al present novel mutations in the genes of NRP1 and NRP2 linked to lymphedema. Although this data suggests a link between these mutations and lymphedema, the authors themselves state ‘‘that these results are not sufficient to associate the two genes, known to be involved in the development of lymphatic capillaries and valves, with lymphedema‘‘. With the present data, it is not clear whether the mutations presented in this study really affect the function of the Nrp1 and Nrp2 proteins. In my opinion, this is true and the authors should at least include an analysis of the functionality of the variant NRP1 and NRP2 proteins in their study for example using biochemistry or cell culture assays. This is expecially important as the described mutations are heterozygous mutations causing single amino acid substitutions. Moreover, the authors should also investigate whether the proteins could have a dominant negative effect.
Author Response
Dear reviewer,
Thank you for your comment. The information requested would certainly be very informative and could improve the manuscript, however at the present time, we are unable obtain such results. We have included a sentence in the Discussion mentioning this limit of the study.
Best regards
Round 2
Reviewer 2 Report
In this revised version of the manuscript Michelini et al. modified the discussion adding future research to perform. However, as crucial experiments verifying the effect of the mutations on the activity of NRP1 and NRP2 proteins are still lacking, these results are still not sufficient to associate the two genes to lymphedema.
Author Response
Dear reviewer,
Thank you for your review. We have removed all reference to the possibility of including detection of these variants in the diagnostic algorithm for lymphedema.
Best regards,